# Application of Sludge-Based Activated Carbons for the Effective Adsorption of Neonicotinoid Pesticides

Eva Sanz-Santos *, Silvia Álvarez-Torrellas *, Lucía Ceballos, Marcos Larriba [ID], V. Ismael Águeda and Juan García [ID]

Catalysis and Separation Processes Group (CyPS), Chemical Engineering and Materials Department, Faculty of Chemistry Sciences, Complutense University, Avda. Complutense s/n, 28040 Madrid, Spain; luciceba@ucm.es (L.C.); marcoslarriba@ucm.es (M.L.); viam@ucm.es (V.I.Á.); jgarciar@ucm.es (J.G.)
* Correspondence: evsanz08@ucm.es (E.S.-S.); satorrellas@ucm.es (S.Á.-T.);
  Tel.: +34-91-394-4114 (E.S.-S.); +34-91-394-4118 (S.Á.-T.)

**Abstract:** The amount of sludge produced in wastewater treatment plants (WWTPs) has increased over the years, and the methods used to reduce this waste, such as incineration, agricultural use, or disposal in landfills, cause problems of secondary pollution. For this reason, it is necessary to find sustainable and low-cost solutions to manage this waste. Additionally, emerging and priority pollutants are attracting attention from the scientific community as they can generate health problems due to inadequate removal in conventional WWTPs. In this work, a pharmaceutical industry sludge was used as a precursor in the synthesis of four activated carbons (ACs) using different activating agents ($ZnCl_2$, $FeCl_3 \cdot 6H_2O$, $Fe(NO_3)_3 \cdot 9H_2O$, and $Fe(SO_4)_3 \cdot H_2O$), to be used for the removal by adsorption of three neonicotinoid pesticides included in latest EU Watch List (Decision 2018/840): acetamiprid (ACT), thiamethoxam (THM), and imidacloprid (IMD). The prepared ACs showed micro–mesoporous properties, obtaining relatively slow adsorption kinetics to reach equilibrium, but despite this, high values of adsorption capacity ($q_e$) were obtained. For example, for AC-$ZnCl_2$ ($S_{BET} = 558$ m$^2$/g), high adsorption capacities of $q_e = 128.9$, $126.8$, and $166.1$ mg/g for ACT, THM, and IMD, respectively, were found. In most cases, the adsorption isotherms showed a multilayer profile, indicating an important contribution of the mesoporosity of the activated carbons in the adsorption process.

**Keywords:** activated carbon; adsorption; neonicotinoid pesticides; sludge-based adsorbents

## 1. Introduction

Sewage sludge is defined as the unavoidable waste generated in wastewater treatment processes and consists of a highly heterogeneous mixture of water and solids. The solid phase consists mainly of heavy metals, organic pollutants, and pathogenic microorganisms [1]. The composition of the sludge varies considerably depending on its source, and it can therefore be classified mainly into two types: if it comes from domestic wastewater, it is known as urban sludge, and if, on the other hand, it comes from industrial water, it is known as industrial sludge. This latter is notable for its high organic matter content and is generally the most suitable precursor to produce carbonaceous adsorbents [2,3].

In the EU, around 13.0 million tons of sewage sludge (in dry matter) are produced annually [4]. The amount of sludge produced has increased enormously with the development of industrialization and urbanization [5,6], and it is considered to be a highly alarming wastewater treatment problem. Different methods have been selected for the removal of sewage sludge, such as disposal in landfills, incineration, and application in agriculture as a fertilizer; but each of these options causes problems of secondary pollution [7,8]. Furthermore, due to increasingly strict legislation (Directive 2008/98/EC) [9,10], sustainable solutions need to be found to manage this waste at low cost. As an alternative, a valorization route has been proposed within the concept of a circular economy; this route consists of using the sludge as a low-cost raw material for its transformation into

high-added-value products such as activated carbons [1,7]—porous carbonaceous materials with adsorbent properties—which have proven to be highly effective in removing a wide range of organic pollutants from water [11–13].

Additionally, the presence of emerging and priority pollutants in the aquatic environment represents an environmental problem of increasing concern. These compounds have low biodegradability, which makes it difficult to eliminate them in wastewater treatment plants (WWTPs) and leads to their frequent presence in the treated effluents. Given their continuous introduction into the environment, micropollutants have been detected in surface water, groundwater, soil, and even in drinking water [14]. Although they appear at low concentrations, their high toxicity, carcinogenic nature, and high persistence in ecosystems mean that they must be considered a great threat to the environment and health [15].

Since 2013, the European Union has begun to monitor and evaluate the risks that these pollutants present, being included in the list of priority pollutants ("Watch List") [16]. In this context, five new neonicotinoid pesticides—thiacloprid (THC), imidacloprid (IMD), thiamethoxam (THM), acetamiprid (ACT), and clothianidin (CLT)—were included in Decision 2018/840.

Despite their exceptional insecticidal properties, the commercial success of these substances has been affected in recent years by the negative effect of neonicotinoids on bees [17], as the role of these animals in pollination is crucial. This environmental situation has had a great social impact, with hundreds of news articles appearing in the media. In addition to the unacceptable risk to bees, in recent years, the negative impact of these pesticides on aquatic ecosystems, given their strong photostability, high solubility, and persistence in the aquatic environment, has been also demonstrated [18].

In the last decade, the advanced tertiary treatments implemented in wastewater treatment plants (WWTPs) have been fundamentally focused on the regulation of macroscopic parameters (suspended solids and turbidity) and microbiological parameters (intestinal nematodes), but without any mention of the presence of emerging and priority pollutants. Therefore, current technologies have not been optimized for the elimination of micropollutants, requiring more specific research in this regard. Several technologies have been proposed and researched to eliminate emerging contaminants from wastewater, including coagulation–flocculation [19] and advanced oxidation processes (AOPs) such as ozonation [20], photocatalysis [21], wet air oxidation [22], and wet air oxidation promoted by hydrogen peroxide [23]. However, processes based on the adsorption of micropollutants onto activated carbon have been presented as the most interesting alternative in WWTPs, since this operation is characterized by its simplicity, low cost, and the non-generation of polluting by-products [24–26].

## 2. Materials and Methods

### 2.1. Materials

The three tested neonicotinoid pesticides (ACT, THM, and IDM), each with a purity of more than 98%, were provided by Sigma-Aldrich and used directly in the experiments. Pesticide solutions were prepared and used in all the experiments with ultrapure water. Hydrochloric acid (HCl, 37 wt.%) was purchased from Honeywell Fluka, acetonitrile (HPLC Plus gradient grade) from Fisher Chemical, and glacial acetic acid from Fluorochem. $ZnCl_2$, $Fe_2(SO_4)_3 \cdot H_2O$, and $Fe(NO_3)_3 \cdot 9H_2O$ reagents were supplied by Sigma-Aldrich, while $FeCl_3 \cdot 6H_2O$ was provided by Panreac AppliChem. Finally, the industrial sludge was supplied by a local pharmaceutical company (Ercros, S.A.) located in Aranjuez (Madrid, Spain).

### 2.2. Preparation of the Carbon Materials

The activated carbons were prepared following the procedure described by Gong [27], with some modifications. Firstly, the sludge was dried in an oven for 24 h at 105 °C. Then, it was ground to a fine powder, and the chemical activation was subsequently carried out; for this purpose, the dried sludge powder was soaked using 4 different activating agents—$ZnCl_2$, $FeCl_3 \cdot 6H_2O$, $Fe(NO_3)_3 \cdot 9H_2O$, and $Fe_2(SO_4)_3 \cdot H_2O$—for a duration of 24 h

at room temperature, followed by drying in an oven at 105 °C for 24 h. Following the activation process, the material was pyrolyzed in a vertical quartz reactor at 800 °C for 2 h, using a $N_2$ flow rate of 100 mL/min and a heating rate of 10 °C/min. Afterward, the activated carbon was washed first with a 5 M HCl solution and then with deionized water. The solid was again dried in an oven at 105 °C for 24 h to continue with a second pyrolysis using the same conditions as described previously. Finally, the resulting activated carbons were ground and sieved (Ø = 50–250 μm).

### 2.3. Characterization of the Sludge

In this work, the part of interest of the sludge was the solid fraction, from which the activated carbons were synthesized. Thus, the chemical oxygen demand (COD) of the aqueous fraction was determined using a PF-11 Macherey-Nagel photometer (Düren, Germany). The total (TS), fixed (FS), and volatile (VS) solids of the sludge were measured according to the Standard Methods [28]. The elemental analysis of the solid was carried out in a LECO CHNS-932 analyzer (Mönchengladbach, Germany), while the chemical composition was determined by X-ray fluorescence spectroscopy (XRF) using PANanalytical Axios equipment (Malvern, UK).

### 2.4. Characterization of the Activated Carbons

The textural properties of the activated carbons were determined by $N_2$ adsorption–desorption at 77 K in a Micromeritics ASAP 2020 system. In the same way, the surface chemistry properties were determined by Fourier transform infrared spectroscopy (FT-IR) using a Thermo Nicolet AVATAR 360 spectrophotometer in the wavenumber range of 400 to 4000 $cm^{-1}$. The thermogravimetric studies of the solids (TGA) were carried out in a PerkinElmer STAR 6000 device, using a temperature range of 25 to 900 °C, a heating rate of 10 °C/min, and a nitrogen volumetric flow rate of 50 mL/min. The elemental analyses were performed in a LECO CHNS-932 analyzer. Finally, the isoelectric point ($pH_{IEP}$) of the samples was measured by zeta potential measurements of a dispersion of 0.05 g of carbon (Ø = 10–20 μm) in 20 mL of deionized water using a Zetasizer Nano ZS (Malvern Instruments, Ltd., Malvern, UK).

### 2.5. Batch Adsorption Studies

Batch adsorption tests were carried out using a LabMate orbital shaker at constant temperature (25 °C) and a shaking rate of 250 rpm. For the kinetic studies, 7.5 mg of adsorbent and 25 mL of each pesticide solution (carbon dose of 0.3 g/L) were maintained under constant agitation. Samples were collected at the determined time intervals until the equilibrium time was reached. The initial adsorbate concentration used in the experiments was 50 mg/L. All the equilibrium adsorption studies were carried out under the same operating conditions, but varying the mass of adsorbent between 1.5 and 37.5 mg (carbon dose ranging from 0.06 to 1.5 g/L). When the equilibrium time was reached, the equilibrium adsorption capacity, $q_e$ (mg/g), was calculated using Equation (1):

$$q_e = \frac{(C_0 - C_e)}{W} \cdot V \tag{1}$$

where $q_e$ (mg/g) is the equilibrium adsorption capacity, $C_0$ (mg/L) is the initial pesticide concentration, $C_e$ (mg/L) is the pesticide concentration at equilibrium, $V$ (L) is the volume of solution, and $W$ (g) the weight of adsorbent.

### 2.6. Analytical Procedure

The pesticide concentration in aqueous samples was analyzed by a high-performance liquid chromatography (HPLC) technique, using Varian ProStar equipment with a "diode array" detector. A Teknokroma column (25 mm × 0.46 mm; 5 μm) was used as the stationary phase, and as the mobile phase, a 70–30% (*v/v*) acetonitrile/75 mM acetic

acid solution was used. The THM, IMD, and ACT concentrations were determined at a wavelength of 270 nm, using a flow rate of 0.8 mL/min.

## 3. Results and Discussion

### 3.1. Characterization of the Sludge

The macroscopic properties of the sludge are reported in Table 1. It is worth highlighting the high COD value of the aqueous fraction (55.36 g/L), which is to be expected since it is an industrial sludge. In addition, both the COD and volatile solids values indirectly indicate the organic matter content of the waste; approximately 80% of the total solids are volatile. These solids are the most relevant, as the greater the amount of organic matter present in the precursor, the higher the efficiency obtained in the synthesis of activated carbon [29]. Another noteworthy value is the Ca content (7.25%), as this is a value that stands out when compared to the rest of the metals present in the sludge. This may be due not only to the origin of the sludge but also to the products used in the wastewater treatments applied in the industry, involving the use of calcium hydroxide, commonly known as slaked lime, which is the most-used calcium-based product in WWTPs.

**Table 1.** Characterization results of the sludge.

| Elemental Analysis (%) | | | | | | |
|---|---|---|---|---|---|---|
| C | O | H | N | P | S | Cl |
| 47.89 | 6.84 | 6.62 | 3.38 | 1.18 | 0.49 | 1.13 |

| Metal Content (%) | | | | | | | | | | | |
|---|---|---|---|---|---|---|---|---|---|---|---|
| Na | Mg | Al | Si | K | Ca | Ti | Mn | Fe | Cu | Zn | Ni |
| 0.15 | 0.02 | 0.38 | 0.17 | 0.49 | 7.25 | 0.02 | 0.01 | 0.19 | 0.01 | 0.02 | 0.03 |

| | |
|---|---|
| Total solids (g/L) | 56.76 |
| Volatile solids (g/L) | 46.24 |
| Fixed solids (g/L) | 10.52 |
| Chemical oxygen demand (COD) (g $O_2$/L) | 55.36 |

### 3.2. Characterization of the Activated Carbons

The textural properties of the synthesized activated carbons were studied through $N_2$ adsorption–desorption analysis. The isotherms and pore size distributions of these materials are shown in Figure 1a,b. The textural parameters, specific surface area ($S_{BET}$), external surface area ($S_{ext}$), total pore volume ($V_{Total}$), and micropore volume ($V_{Micro}$) are reported in Table 2.

**Table 2.** Textural properties of the carbon materials.

| Adsorbent | $S_{BET}$ (m²/g) | $S_{ext}$ (m²/g) | $V_{Total}$ (cm³/g) | $V_{Micro}$ (cm³/g) | $V_{Micro}/V_{Total}$ |
|---|---|---|---|---|---|
| AC-ZnCl₂ | 558 | 145 | 0.35 | 0.15 | 0.43 |
| AC-FeCl₃ | 468 | 170 | 0.56 | 0.16 | 0.29 |
| AC-Fe(NO₃)₃ | 240 | 158 | 0.32 | 0.04 | 0.13 |
| AC-Fe₂(SO₄)₃ | 233 | 203 | 0.48 | 0.01 | 0.02 |

According to the International Union of Pure and Applied Chemistry (IUPAC) classification, all the samples exhibited type IV isotherms, characteristic of mainly mesoporous materials. At medium pressure, due to capillary condensation, the desorption branch did not match with the adsorption isotherm, causing a hysteresis loop, which demonstrates the presence of wider pores (meso- and macropores) in the carbon structure [30]. In addition, the hysteresis loops presented by the activated carbons showed two parallel and

almost horizontal branches (except for the case of AC-Fe$_2$(SO$_4$)$_3$), which can be classified as H4-type hysteresis loops, according to the IUPAC classification.

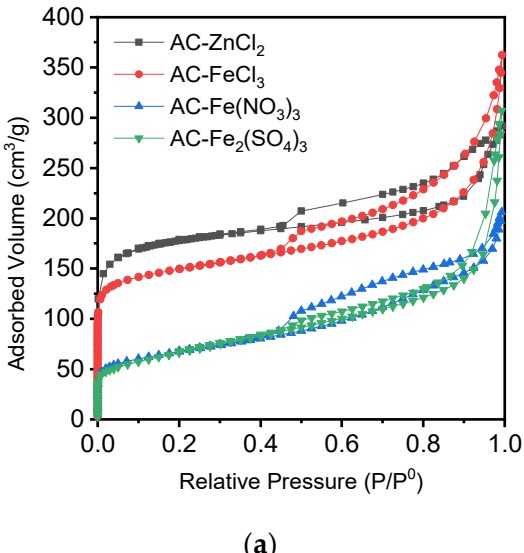
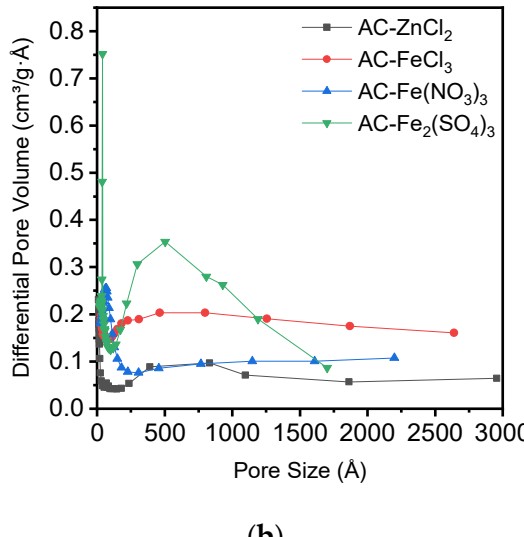

(**a**)                                           (**b**)

**Figure 1.** N$_2$ adsorption–desorption isotherms (**a**) and pore size distributions (**b**) of the activated carbon (AC) materials.

Sludge is a heterogeneous waste and, therefore, generates activated carbons with heterogeneous textural properties—that is, with meso–microporous character. This phenomenon was previously observed in activated carbons synthesized from biomass [31,32]. In the case of AC-ZnCl$_2$, it can be observed that part of the pore filling occurs at low pressure, since it could be considered a mainly microporous activated carbon. In contrast, in the case of AC-Fe$_2$(SO$_4$)$_3$ carbon, a significant contribution of macroporosity was observed, showing capillary condensation at high pressure values.

The pore size distributions of the materials were evaluated by the Barrett–Joyner–Halenda (BJH) method. It can be seen that all the samples showed a pore distribution mainly centered in the micro–mesoporosity range, except for the AC-Fe$_2$(SO$_4$)$_3$ sample, which showed wider pores (~50 nm).

Table 2 shows that the AC-ZnCl$_2$ carbon showed the highest specific surface area. The order according to development of the specific surface area for the different synthesized carbons was as follows: AC-ZnCl$_2$ > AC-FeCl$_3$ > AC-Fe(NO$_3$)$_3$ > AC-Fe$_2$(SO$_4$)$_3$. Regarding the pore volume, AC-Fe$_2$(SO$_4$)$_3$ carbon showed a practically negligible micropore volume, whereas with the AC-ZnCl$_2$ sample, the opposite was observed, i.e., the micropore volume corresponded to practically half of the total pore volume. This behavior was corroborated by the N$_2$ adsorption–desorption isotherms, as discussed before.

The FT-IR spectra of the activated carbons are depicted in Figure 2a. A broad band located at ~3400 cm$^{-1}$ can be attributed to O–H stretching vibration due to the presence of water in the carbon samples. The band close to 2900 cm$^{-1}$, very strong for the AC-ZnCl$_2$ and AC-FeCl$_3$ samples, is associated with stretching vibration of aliphatic C–H bonds, whereas the peak at 2331 cm$^{-1}$, observed only in the AC-Fe(NO$_3$)$_3$ and AC-Fe$_2$(SO$_4$)$_3$ spectra, indicates the presence of ketone functional groups. The band close to 1560 cm$^{-1}$ corresponds to C=C stretching vibration of the aromatic rings [33]. Thus, the bands associated with alcohols, phenols, esters, acids, and ethers were observed at a wavelength of 1100 cm$^{-1}$. Finally, the bands attributed to out-of-plane bending vibrations of O–H and C–H were observed between 500 and 750 cm$^{-1}$ [34].

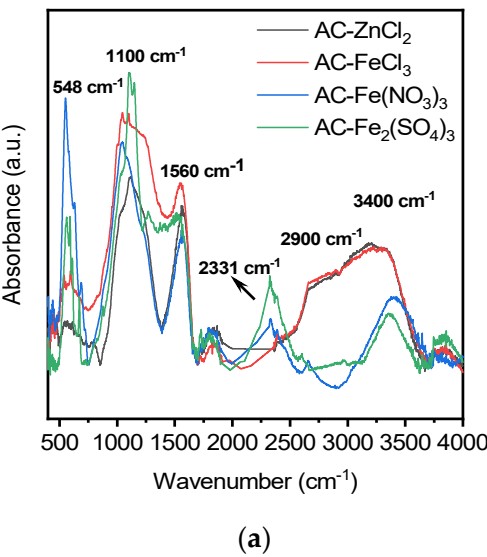

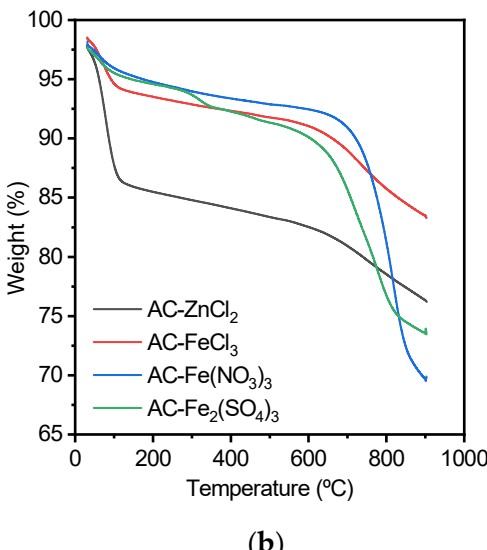

**(a)**          **(b)**

**Figure 2.** FT-IR spectra (**a**) and thermogravimetric analysis (**b**) of the carbon materials.

Figure 2b shows the thermogravimetric (TG) analysis results of the synthesized activated carbons, which evaluated the thermal stability of the solids. In all the samples, a weight loss was observed at temperatures around 100 °C, which is attributed to the evaporation of the adsorbed water. The weight loss observed between 200 and 400 °C may be due to the decomposition of volatile organic and inorganic compounds, and that in the range between 400 and 600 °C may be attributed to the decomposition of high-molecular-weight compounds. At temperatures around 600 °C, a more abrupt weight loss was observed for activated carbons synthesized with $Fe(NO_3)_3$ and $Fe_2(SO_4)_3$, whereas this drop was more stable for activated carbons synthesized from impregnating agents with chloride, i.e., $ZnCl_2$ and $FeCl_3$. Therefore, the weight loss observed at 600 °C can be mainly attributed to the thermal decomposition of the used activating agent.

The elemental analysis results of the samples and the corresponding isoelectric point ($pH_{PIE}$) values are collected in Table 3. Generally, activated carbons are mainly composed of fixed carbon and inorganic compounds. As can be seen in Table 3, the fixed carbon contents (C, wt.%) were 55.60, 59.77, 33.51, and 31.71% for AC-$ZnCl_2$, AC-$FeCl_3$, AC-$Fe(NO_3)_3$, and AC-$Fe_2(SO_4)_3$, respectively—values significantly lower than those found in activated carbons obtained from fossil fuels [25].

**Table 3.** Elemental analysis results and isoelectric point ($pH_{IEP}$) values of the carbon materials.

| Adsorbent | C (wt.%) | H (wt.%) | N (wt.%) | S (wt.%) | $pH_{IEP}$ |
|---|---|---|---|---|---|
| AC-$ZnCl_2$ | 55.60 | 2.55 | 4.00 | 1.82 | 2.14 |
| AC-$FeCl_3$ | 59.77 | 2.36 | 3.36 | 1.35 | 5.40 |
| AC-$Fe(NO_3)_3$ | 33.51 | 0.99 | 2.05 | 0.66 | 6.81 |
| AC-$Fe_2(SO_4)_3$ | 31.71 | 1.05 | 1.74 | 13.15 | 3.08 |

Even when the material used as a precursor has a high organic matter content, at a high carbonization temperature, the organic matter decomposes; this causes a decrease in the fixed carbon percentage, transforming it into ash. Therefore, the higher the carbonization temperature, the lower the percentage of fixed carbon found in the final material, indicating greater carbonization of organic matter. For this reason, the carbonization temperature of sludge in industry usually varies between 400 and 600 °C, since in this temperature range, the product obtained is often suitable for use as an alternative fuel due to its high organic content [35]. Also, it is worth highlighting the S content of 13.15% measured in AC-$Fe_2(SO_4)_3$, indicating that impregnation with $Fe_2(SO_4)_3$ was efficiently carried out. On the other hand, according to the literature, washing with HCl promotes the production of

acidic adsorbents [36], although it is necessary to correctly remove ash residues that clog the pores of the final materials. Furthermore, independently of the washing conditions, the use of activating agents such as $FeCl_3$ and $ZnCl_2$ [37,38] conditioned the preparation of predominantly acidic carbon materials, as can be observed from their isoelectric point ($pH_{IEP}$) values.

### 3.3. Pesticide Adsorption Studies

### 3.3.1. Kinetic Adsorption Studies

The adsorption kinetics of the pesticides onto the synthesized activated carbons are depicted in Figure 3a–d.

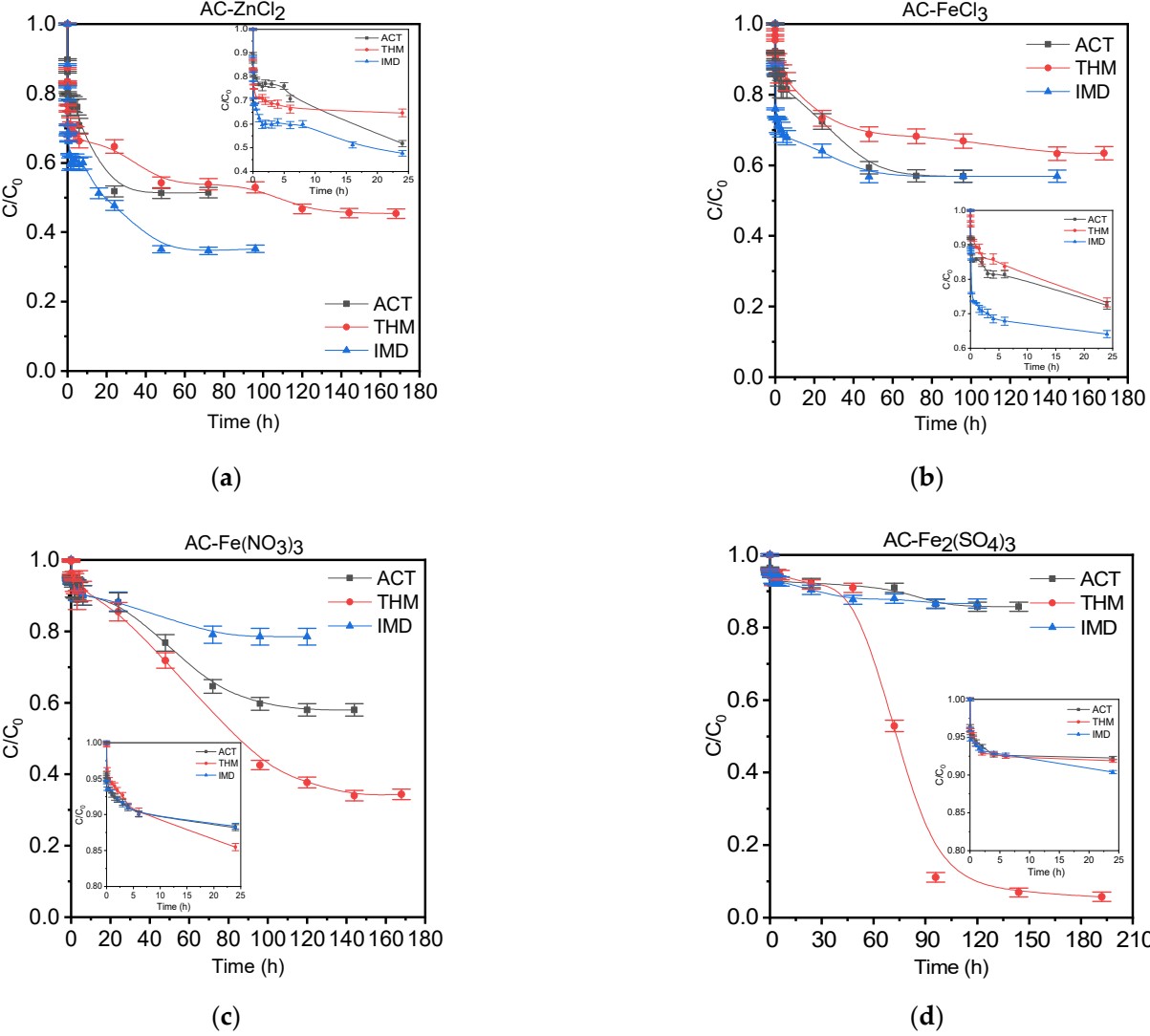

**Figure 3.** Adsorption kinetics of the three pesticides onto the (**a**) AC-$ZnCl_2$, (**b**) AC-$FeCl_3$, (**c**) AC-$Fe(NO_3)_3$, and (**d**) AC-$Fe_2(SO_4)_3$ activated carbons.

Slow adsorption rates were observed for the three pesticides, achieving equilibrium times ranging from 24 to 144 h. As can be seen in Figure 3, in the first stage, rapid adsorption occurred because adsorption sites on the surface are easily accessed and occupied by the micropollutant; in the next stage, the diffusional effect comes into play (conditioned by the steric hindrance caused by the adsorbate molecules), due to the relatively high molecular size of the pesticides used compared to the medium available pore size of the activated carbons, requiring in this stage more time to diffuse until reaching equilibrium. According to this, longer equilibrium times were found when the molecular volume of the pesticides

used was larger, following the order ACT (266 Å$^3$) < IMD (271 Å$^3$) < THM (303 Å$^3$). For this reason, the slowest equilibrium time (144 h) was obtained for the larger pesticide, THM.

In the same way, very high adsorption removal values were obtained at equilibrium time. The best results were achieved with the AC-ZnCl$_2$ sample. Generally, it was observed that better adsorption capacities were obtained with those activated carbons that showed a higher specific surface area (Table 2); therefore, it is clear that the textural properties of the materials played a key role in their adsorption performance. An exception was observed for THM removal using the AC-Fe(NO$_3$)$_3$ and AC-Fe$_2$(SO$_4$)$_3$ samples, where a simultaneous contribution of adsorption and reaction phenomena was observed (Figure 3c,d). In both cases, a first stage related to the adsorption process and a second stage where the adsorption process occurred simultaneously with a possible degradation-mediated by-reaction were observed; this phenomenon may be due to the high iron content found in these carbonaceous materials.

Pseudo-first-order and pseudo-second-order kinetic models were applied to assess the experimental kinetic data. The pseudo-first-order (PPO) kinetic model, also known as the Lagergren equation, assumes that physisorption limits the adsorption rate on the adsorbent particles and can be expressed by the following equation:

$$\ln(q_e - q) = \ln q_e - k_1 \cdot t \tag{2}$$

where $k_1$ (h$^{-1}$) is the first-order reaction rate equilibrium constant, and $q$ (mg/g) and $q_e$ (mg/g) denote the adsorption capacity at any time and at the equilibrium time, respectively.

On the other hand, the pseudo-second-order (PSO) kinetic model assumes that chemisorption is the rate-limiting step of the adsorption process, and it is expressed as follows:

$$\frac{t}{q} = \frac{1}{k_2 \cdot q_e^2} + \frac{t}{q_e} \tag{3}$$

where $k_2$ (g/mg·h) is the rate constant of the pseudo-second-order kinetic model.

As shown in Table 4, considering the experimental and estimated adsorption capacities and R$^2$ values, both the pseudo-first-order and pseudo-second-order kinetic models well described the adsorption of pesticides onto the synthesized activated carbons. This suggests that physical and chemical interactions simultaneously contribute to and control the adsorption of the pesticides on the surfaces of the activated carbons employed. Generally, physical adsorption, or physisorption, involves van der Waals forces, while chemical adsorption, or chemisorption, occurs when electrons are shared or transferred through covalent bonds [39].

### 3.3.2. Isotherm Adsorption Studies

The adsorption isotherms of the three studied pesticides onto the synthesized adsorbents are shown in Figure 4a–d. The initial concentration of micropollutant was fixed at 50 mg/L, whereas the dose of adsorbent was varied within the range of 0.05–1.5 mg/mL. From the results, the best adsorption capacities for the three contaminants were found for AC-ZnCl$_2$: q$_e$ = 128.9, 126.8, and 166.1 for ACT, THM, and IMD, respectively. As can be observed, all the adsorption isotherms showed a multilayer profile. Thus, they could be classified as type S-3 and S-4, according to the classification established by Giles et al. [40]. The S-type isotherms are characterized by a steep slope, and as the adsorbate concentration increases, the adsorption capacity approaches a first plateau. The first plateau is attributed to saturation caused by the termination of the first monolayer. Then, as the adsorbate concentration increases, a vertical rearrangement of the adsorbed molecules occurs, and more active sites become available for adsorption [41]. Therefore, the newly available adsorption sites lead to a further increase of the adsorption capacity, and a second plateau occurs, which is attributed to a second layer. These S-type isotherms may also be the result of a competitive effect between the solvent and the adsorbate towards the active sites available for adsorption [42].

**Table 4.** Kinetic model parameters for the adsorption of ACT, THM, and IMD onto the carbon materials.

| Adsorbent | Pesticide | $q_{exp}$ (mg/g) | Pseudo-First-Order Model | | | Pseudo-Second-Order Model | | |
|---|---|---|---|---|---|---|---|---|
| | | | $q_{teor}$ (mg/g) | $k_1$ ($h^{-1}$) | $R^2$ | $q_{teor}$ (mg/g) | $k_2 \times 1^{-2}$ (g/mg·h) | $R^2$ |
| **AC-ZnCl$_2$** | **ACT** | 84.3 | 84.2 | 0.28 | 0.870 | 78.8 | 0.71 | 0.867 |
| | **THM** | 96.9 | 96.9 | 1.39 | 0.804 | 96.7 | 3.50 | 0.840 |
| | **IMD** | 105.7 | 105.7 | 1.37 | 0.824 | 105.0 | 2.95 | 0.861 |
| **AC-FeCl$_3$** | **ACT** | 70.1 | 70.0 | 0.17 | 0.929 | 65.0 | 0.39 | 0.931 |
| | **THM** | 66.1 | 66.1 | 0.16 | 0.968 | 93.8 | 1.08 | 0.863 |
| | **IMD** | 75.9 | 64.3 | 0.02 | 0.943 | 59.0 | 70.2 | 0.963 |
| **AC-Fe(NO$_3$)$_3$** | **ACT** | 66.4 | 60.4 | 0.02 | 0.967 | 55.6 | 0.80 | 0.941 |
| | **THM** | 135.0 | 124.5 | 0.02 | 0.984 | 111.9 | 0.02 | 0.964 |
| | **IMD** | 35.4 | 35.4 | 0.13 | 0.887 | 33.0 | 0.55 | 0.899 |
| **AC-Fe$_2$(SO$_4$)$_3$** | **ACT** | 23.3 | 23.3 | 0.28 | 0.853 | 39.8 | 29.47 | 0.933 |
| | **THM** | 194.8 | 156.1 | 0.01 | 0.912 | 138.7 | 0.01 | 0.904 |
| | **IMD** | 21.5 | 21.5 | 0.26 | 0.888 | 64.4 | 2.55 | 0.946 |

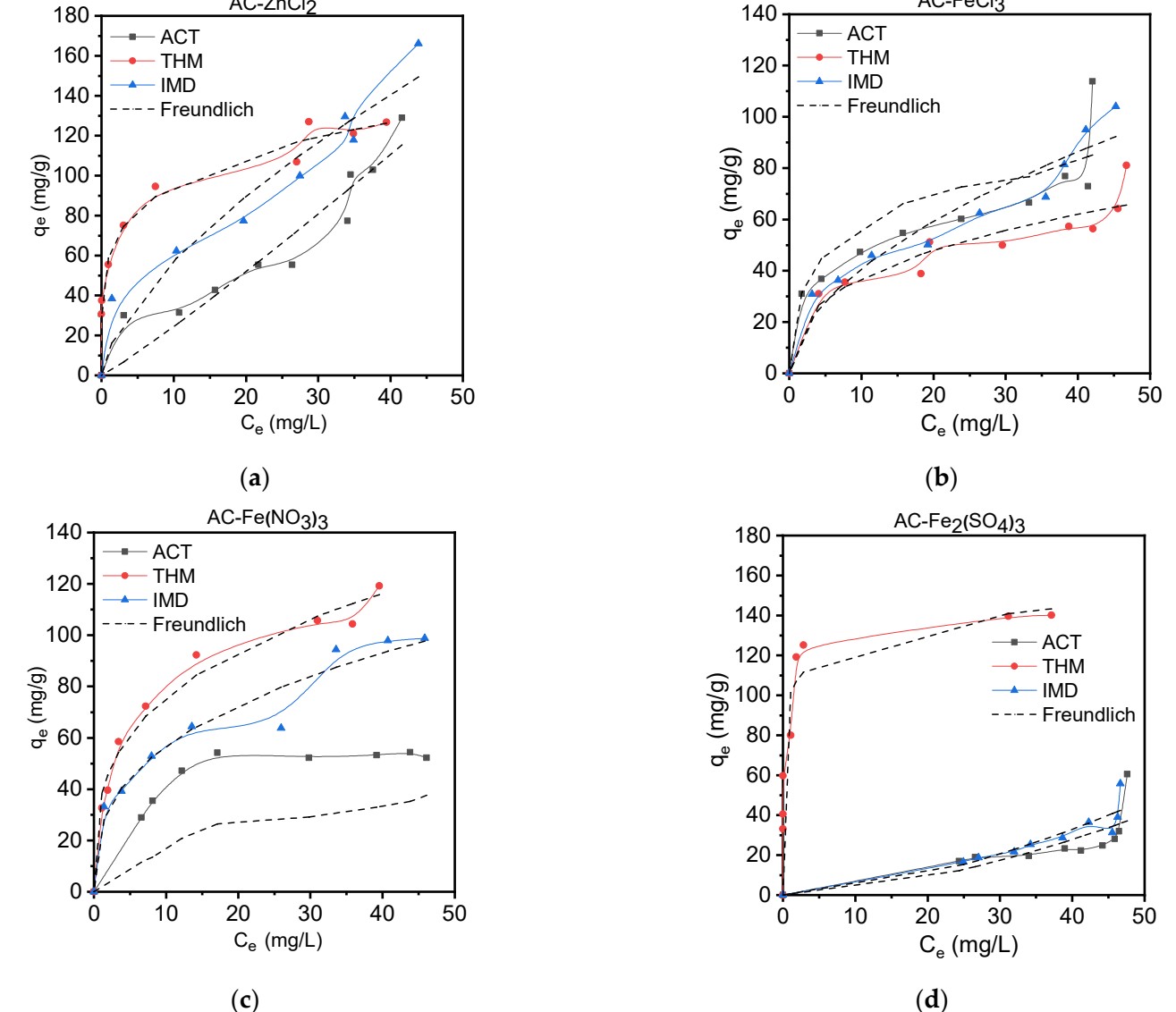

**Figure 4.** Adsorption isotherms of the three pesticides onto the (**a**) AC-ZnCl$_2$, (**b**) AC-FeCl$_3$, (**c**) AC-Fe(NO$_3$)$_3$, and (**d**) AC-Fe$_2$(SO$_4$)$_3$ activated carbons.

Correlation of the experimental equilibrium data to empirical models is essential for the practical interpretation of scientific results. In this work, the Langmuir, Freundlich, Sips, and Guggenheim–Anderson–de Boer (GAB) isotherm models were used to fit the experimental adsorption data. The Langmuir isotherm model supposes that the adsorption process takes place in a monolayer, that the solid surface is energetically homogeneous, and that the molecules are adsorbed on well-defined sites, i.e., without migration of the adsorbate from one active site to another [43]. This model is expressed by the following equation:

$$q_e = \frac{q_{sat} \cdot b \cdot C_e}{1 + b \cdot C_e} \tag{4}$$

where $q_e$ is the equilibrium adsorption capacity (mg/g), $C_e$ is the equilibrium concentration (mg/L), $q_{sat}$ is the maximum adsorption capacity in the monolayer (mg/g), and $b$ is a constant related to the affinity between the adsorbate and adsorbent (mg/L).

The Freundlich isotherm is an empirical model used when the adsorption mechanism appears to be mixed and the material surface can be considered energetically heterogeneous. As a consequence, this model can be applied to multilayer adsorption isotherms [44]. The Freundlich equation is as follows:

$$q_e = K_F \cdot C_e^{1/n_f} \tag{5}$$

where $q_e$ is the equilibrium adsorption capacity (mg/g), $k_f$ (L/g) is the Freundlich constant related to the adsorption capacity of the adsorbent, and $1/n_f$ indicates the intensity of the adsorption process or the surface heterogeneity; when the value of $n_f$ is >1, the adsorption conditions can be considered as favorable.

The Sips model was developed to give the Freundlich model a finite limit at high equilibrium concentrations. The difference from the Langmuir isotherm model is an additional parameter, $n$; when $n = 1$, the Sips model simplifies to the Langmuir equation. The parameter $n$ can be defined as a measure of the surface heterogeneity: this heterogeneity may be due to the adsorbent surface, the adsorbate, or both. The value of $n$ is usually higher than unity, and the greater the value, the more heterogeneous the system. The Sips model can be expressed by the following equation:

$$q_e = \frac{q_{sat} \cdot (b \cdot C_e)^{1/n}}{1 + (b \cdot C_e)^{1/n}} \tag{6}$$

where $b$ (L/mg) and $1/n$ are temperature-dependent Sips parameters, and $q_{sat}$ (mg/g) can be taken as a function (or not) of temperature.

Finally, the Guggenheim–Anderson–De Boer (GAB) isotherm model is an extension of the Langmuir isotherm model that considers multilayer adsorption by assuming that each adsorbed molecule provides a site for the second one. The expression of the GAB model is as follows, assuming that only two layers are formed in the adsorption process:

$$q_e = \frac{q_m \cdot K_1 \cdot C_e}{(1 - K_2 \cdot C_e) \cdot [1 + (K_1 - K_2) \cdot C_e]} \tag{7}$$

where $K_1$ and $K_2$ (L/mg) are equilibrium constants attributable to the first and second layer, respectively, and $q_m$ (mg/g) is the maximum adsorption capacity in the first layer.

The isotherm model parameters for adsorption onto the synthesized activated carbons are shown in Tables 5 and 6. The best fit to the experimental data of the adsorption isotherms was found for the Freundlich model, with higher values of the correlation coefficient ($R^2$); this behavior was expected since it is a model suitable for fitting heterogeneous adsorption systems with an exponential distribution of active sites [45].

**Table 5.** Isotherm model parameters for the adsorption of ACT, THM, and IMD onto the carbon materials: the Langmuir and Freundlich models.

| Sample | Pesticide | Langmuir | | | | | Freundlich | | | |
|---|---|---|---|---|---|---|---|---|---|---|
| | | $q_{exp}$ (mg/g) | $q_{teor}$ (mg/g) | $q_{sat} \times 10^{-5}$ (mg/g) | b (L/mg) | $R^2$ | $q_{teor}$ (mg/g) | $K_F$ (L/g) | $n_f$ | $R^2$ |
| AC-ZnCl$_2$ | ACT | 128.9 | 112.9 | 8.86 | 3.1 | 0.955 | 115.3 | 1.98 | 0.9 | 0.959 |
| | THM | 126.8 | 149.5 | 8.99 | 4.2 | 0.875 | 126.3 | 59.16 | 4.9 | 0.978 |
| | IMD | 166.1 | 164.6 | 8.85 | 4.2 | 0.975 | 149.4 | 12.82 | 1.5 | 0.976 |
| AC-FeCl$_3$ | ACT | 113.9 | 95.9 | 8.85 | 2.6 | 0.889 | 85.1 | 16.24 | 2.3 | 0.922 |
| | THM | 81.1 | 82.2 | 8.85 | 1.7 | 0.815 | 65.6 | 15.81 | 2.7 | 0.950 |
| | IMD | 104.1 | 103.6 | 8.85 | 2.6 | 0.960 | 92.3 | 11.71 | 1.9 | 0.969 |
| AC-Fe(NO$_3$)$_3$ | ACT | 60.6 | 34.2 | 8.87 | 0.8 | 0.768 | 37.6 | 0.04 | 0.6 | 0.793 |
| | THM | 140.2 | 160.5 | 8.99 | 4.8 | 0.679 | 143.3 | 100.95 | 10.3 | 0.931 |
| | IMD | 55.8 | 39.0 | 8.85 | 0.9 | 0.899 | 42.6 | 0.06 | 0.6 | 0.926 |
| AC-Fe$_2$(SO$_4$)$_3$ | ACT | 60.6 | 34.2 | 8.85 | 0.8 | 0.768 | 37.1 | 0.06 | 0.6 | 0.791 |
| | THM | 140.2 | 160.5 | 8.99 | 4.8 | 0.678 | 143.3 | 100.95 | 10.3 | 0.931 |
| | IMD | 55.8 | 38.7 | 8.85 | 0.9 | 0.899 | 42.4 | 0.08 | 0.6 | 0.925 |

**Table 6.** Isotherm model parameters for the adsorption of ACT, THM, and IMD onto the carbon materials: the Sips and Guggenheim–Anderson–De Boer (GAB) models.

| Sample | Pesticide | Sips | | | | | | GAB | | | | |
|---|---|---|---|---|---|---|---|---|---|---|---|---|
| | | $q_{exp}$ (mg/g) | $q_{teor}$ (mg/g) | $q_{sat}$ (mg/g) | $b \times 10^5$ (L/mg) | n | $R^2$ | $q_{teor}$ (mg/g) | $q_m \times 10^{-5}$ (mg/g) | $K_1 \times 10^6$ (L/mg) | $K_2 \times 10^6$ (L/mg) | $R^2$ |
| AC-ZnCl$_2$ | ACT | 128.9 | 115.3 | 125,557 | 3.96 | 0.9 | 0.958 | 112.9 | 8.87 | 3.06 | 3.06 | 0.955 |
| | THM | 126.8 | 124.9 | 1373 | 0.03 | 4.8 | 0.976 | 150.4 | 8.87 | 4.30 | 3.07 | 0.875 |
| | IMD | 166.1 | 149.3 | 50,683 | 0.29 | 1.5 | 0.972 | 164.6 | 8.87 | 4.23 | 3.07 | 0.972 |
| AC-FeCl$_3$ | ACT | 113.9 | 83.7 | 2538 | 0.52 | 2.4 | 0.866 | 95.9 | 8.87 | 2.57 | 3.06 | 0.894 |
| | THM | 81.1 | 65.9 | 1927 | 0.31 | 2.6 | 0.949 | 75.7 | 8.87 | 1.83 | 3.06 | 0.898 |
| | IMD | 104.1 | 91.8 | 1360 | 20.30 | 1.7 | 0.968 | 103.6 | 8.87 | 2.58 | 3.06 | 0.960 |
| AC-Fe(NO$_3$)$_3$ | ACT | 60.5 | 37.2 | 243,432 | 9.57 | 0.6 | 0.791 | 34,.2 | 8.87 | 0.81 | 3.06 | 0.768 |
| | THM | 140.2 | 159.5 | 1378 | 0.12 | 4.8 | 0.929 | 160.5 | 8.87 | 4.88 | 3.07 | 0.679 |
| | IMD | 55.8 | 42.7 | 285,175 | 11.70 | 0.5 | 0.925 | 39.0 | 8.87 | 0.94 | 3.06 | 0.899 |
| AC-Fe$_2$(SO$_4$)$_3$ | ACT | 60.5 | 37.9 | 3740 | 169.1 | 0.5 | 0.793 | 34.2 | 8.87 | 8.10 | 3.06 | 0.768 |
| | THM | 140.2 | 139.1 | 139 | 108,617 | 0.4 | 0.950 | 160.5 | 8.87 | 4.88 | 3.07 | 0.678 |
| | IMD | 55.8 | 42.7 | 7231 | 104.4 | 0.5 | 0.926 | 38.7 | 886,713 | 0.93 | 3.06 | 0.899 |

Also, it is worth noting that for the adsorption onto AC-Fe$_2$(SO$_4$)$_3$, the Sips isotherm model fitted equally as well as Freundlich model. Regarding the parameter $n_f$ of the Freundlich equation, generally, values higher than unity are indicative that significant adsorption takes place at low concentration values (favorable adsorption isotherms), since the increase in the adsorption capacity with aqueous concentration becomes less significant at a higher concentration ($n_f$ values lower than unity). Most of the obtained $n_f$ values were greater than unity, thus verifying good adsorption onto a heterogeneous carbon surface [46,47].

Although the Freundlich isotherm model achieved the best results, it did not provide a complete reproduction of the experimental data, mainly due to the complexity of reproducing multilayer adsorption isotherms and the large plateau region exhibited between the layers.

*3.4. Comparison of Pesticide Adsorption Capacity with That of Other Adsorbents Derived from Biomass Sources*

Table 7 shows the adsorption capacities obtained in previous studies for the adsorption of the pesticides studied in this work, i.e., acetamiprid and imidacloprid, with adsorbents synthesized from different biomass sources.

**Table 7.** Adsorption capacity values of pesticides onto adsorbents derived from biomass.

| Biomass Source | Pesticide | Adsorption Capacity (mg/g) | $S_{BET}$ (m²/g) | $t_e$ (min) | Reference |
|---|---|---|---|---|---|
| *Ricinodendron heudelotii* shells | Imidacloprid | 43.48 | 1179 | 90 | [48] |
| Peach stone | Imidacloprid | 39.37 | 6 | 40 | [49] |
| Tangerine peels | Acetamiprid | 35.70 | 688 | 240 | [50] |
| Peanut shell | Imidacloprid | 8.68 | 535 | 240 | [51] |
| Sugarcane bagasse | Imidacloprid | 313.00 | 660 | 720 | [52] |

The best results in this work were obtained with the AC-ZnCl₂ activated carbon ($S_{BET}$ = 558 m²/g), with adsorption capacities of 128.9, 126.8, and 166.1 mg/g for ACT, IMD, and THM, respectively, at 24 h. When comparing the results obtained for AC-ZnCl₂ with those shown in Table 7, it can be seen that a higher adsorption capacity was obtained with the pesticide IMD when using sugarcane bagasse as a precursor, while for the pesticide ACT, lower adsorption capacity values were obtained; hence, it can be concluded that the adsorption capacity varies considerably depending on the origin of the activated carbon. On the other hand, if $S_{BET}$ values are considered, excluding that for the activated carbon from peach stones, the values were equal to or even higher than that found for the AC-ZnCl₂ activated carbon; however, the adsorption capacity and equilibrium time values were lower, except those for the activated carbon from sugarcane bagasse. Therefore, if the referenced studies did not allow enough contact time between the adsorbent and adsorbate, it is possible that saturation of the activated carbon did not occur, which would explain the very fast equilibrium times. For the activated carbon from sugarcane bagasse, the adsorption capacity was the highest ($q_e$ = 313 mg/g), and it took 720 min to reach the equilibrium. Furthermore, it should be noted that no literature results were found using the pesticide THM, which shows the innovativeness of this work. Furthermore, AC-ZnCl₂ outperformed all previously published adsorbents tested on ACT and outperformed three of four published adsorbents for IMD. In conclusion, more research is needed to determine the performance of this kind of adsorbent in pesticide removal.

## 4. Conclusions

The use of sewage sludge as a low-cost precursor for the preparation of activated carbons to removal neonicotinoid pesticides, e.g., acetamiprid, thiamethoxam, and imidacloprid, is a sustainable way to solve the serious problem of sludge management while being in accordance with the concept known as a circular economy. The synthesized activated carbons showed heterogeneous textural properties (micro–mesoporous character)—a characteristic of activated carbons obtained from waste and perhaps also as a result of the heterogeneous nature of the precursor used. The maximum specific surface area ($S_{BET}$ = 558 m²/g) was obtained for the AC synthesized using ZnCl₂ as the impregnation agent. Relatively slow adsorption kinetics were obtained for all the materials as a result of the steric hindrance caused by the molecules adsorbed, that is, due to the relatively large molecular size of the pesticides in relation to the medium pore size of the activated carbons; thus, the equilibrium times were longer when THM was used, since this is the largest of the tested molecules. Despite this, very high equilibrium adsorption capacities were obtained: using AC-ZnCl₂, $q_e$ values of 128.9, 126.8, and 166.1 mg/g for ACT, THM, and IMD, respectively, were obtained. Besides this, it is important to highlight the behavior observed in the removal of THM onto the AC-Fe(NO₃)₃ and AC-Fe₂(SO₄)₃ activated carbons, since simultaneous contributions of adsorption and reaction phenomena were observed; this is possibly due to the high content in Fe of these materials. Regarding the adsorption isotherms, it is interesting that, in general, they showed a multilayer profile, which indicates an important contribution of the mesoporosity in the adsorption process.

For all these reasons, the use of sewage sludge to produce activated carbons has great potential as it is a sustainable solution for the management of this kind of waste, and the

resulting activated carbons can be considered capable of competing with conventional adsorbent materials synthesized from fossil fuels.

**Author Contributions:** Conceptualization, E.S.-S., S.Á.-T. and M.L.; methodology, E.S.-S. and L.C.; software, E.S.-S. and L.C.; formal analysis, E.S.-S., S.Á.-T. and M.L.; investigation, E.S.-S., L.C. and S.Á.-T.; resources, V.I.Á. and J.G.; data curation, E.S.-S., L.C. and S.Á.-T.; writing—original draft preparation, E.S.-S., L.C. and S.Á.-T.; writing—review and editing, E.S.-S., L.C., S.Á.-T. and M.L.; supervision, V.I.Á. and J.G. All authors have read and agreed to the published version of the manuscript.

**Funding:** This research was funded by the Regional Government of Madrid provided through Project P2018/EMT-4341 and Project IND2019/AMB-17114.

**Institutional Review Board Statement:** Not Applicable.

**Informed Consent Statement:** Not Applicable.

**Data Availability Statement:** The study did not report any data.

**Conflicts of Interest:** The authors declare no potential conflicts of interest with respect to the research, authorship, and/or publication of the article.

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
