# Peer review of "Application of Sludge-Based Activated Carbons for the Effective Adsorption of Neonicotinoid Pesticides"

_applsci, doi:10.3390/app11073087_

Round 1

Reviewer 1 Report

The paper provides an interesting insight into the adsorption performance of a sludge-based activated carbons for the adsorption of neonicotinoid pesticides through batch adsorption tests, but there are main concerns at the present stage of this work that should be addressed before publication. Please see the attached pdf file.

Reviewer 2 Report

Dear Authors 

It was a pleasure for me to read Your paper. I would like to recommend it for publication in Applied Sciences Journal. I do not have any significant comments to the manuscript, it was very well written in terms of science, but I have some comments to the editorial side, though it shall by no means deteriorate the value of the article. The fonts used in the all figures  should be enlarged, since currently the font on the remaining figures is too small. Larger fonts will be friendlier to readers, and therefore will be more likely to read and cited your article.

Yours sincerely
